# Voxel-Wise Comparison of Co-Registered Quantitative CT and Hyperpolarised Gas Diffusion-Weighted MRI Measurements in IPF

**DOI:** 10.3390/diagnostics13233497

**Published:** 2023-11-21

**Authors:** Ho-Fung Chan, Nicholas D. Weatherley, Alberto M. Biancardi, Christopher S. Johns, Bilal A. Tahir, Ronald A. Karwoski, Brian J. Bartholmai, Stephen M. Bianchi, Jim M. Wild

**Affiliations:** 1POLARIS, Imaging Section, Division of Clinical Medicine, University of Sheffield, Sheffield S10 2JF, UK; fung.chan@auckland.ac.nz (H.-F.C.); christopher.johns@nhs.net (C.S.J.); b.tahir@sheffield.ac.uk (B.A.T.); 2Academic Directorate of Respiratory Medicine, Sheffield Teaching Hospitals NHS Foundation Trust, Sheffield S5 7AU, UK; 3Radiology Department, Sheffield Teaching Hospitals NHS Foundation Trust, Sheffield S5 7AU, UK; 4Department of Oncology and Metabolism, University of Sheffield, Sheffield S10 2JF, UK; 5Biomedical Imaging Resource, Mayo Clinic, Rochester, MN 55905, USA; 6Department of Radiology, Mayo Clinic, Rochester, MN 55905, USA; 7Insigneo Institute, University of Sheffield, Sheffield S1 3JD, UK

**Keywords:** idiopathic pulmonary fibrosis, diffusion-weighted MRI, hyperpolarised gas, lung MRI, quantitative lung CT, spatial co-registration

## Abstract

The patterns of idiopathic pulmonary fibrosis (IPF) lung disease that directly correspond to elevated hyperpolarised gas diffusion-weighted (DW) MRI metrics are currently unknown. This study aims to develop a spatial co-registration framework for a voxel-wise comparison of hyperpolarised gas DW-MRI and CALIPER quantitative CT patterns. Sixteen IPF patients underwent ^3^He DW-MRI and CT at baseline, and eleven patients had a 1-year follow-up DW-MRI. Six healthy volunteers underwent ^129^Xe DW-MRI at baseline only. Moreover, ^3^He DW-MRI was indirectly co-registered to CT via spatially aligned ^3^He ventilation and structural ^1^H MRI. A voxel-wise comparison of the overlapping ^3^He apparent diffusion coefficient (ADC) and mean acinar dimension (Lm_D_) maps with CALIPER CT patterns was performed at baseline and after 1 year. The abnormal lung percentage classified with the Lm_D_ value, based on a healthy volunteer ^129^Xe Lm_D_, and CALIPER was compared with a Bland–Altman analysis. The largest DW-MRI metrics were found in the regions classified as honeycombing, and longitudinal DW-MRI changes were observed in the baseline-classified reticular changes and ground-glass opacities regions. A mean bias of −15.3% (95% interval −56.8% to 26.2%) towards CALIPER was observed for the abnormal lung percentage. This suggests DW-MRI may detect microstructural changes in areas of the lung that are determined visibly and quantitatively normal by CT.

## 1. Introduction

Idiopathic pulmonary fibrosis (IPF) is a group of lung diseases that are defined by the lack of an underlying cause and are characterised by the presence of usual interstitial pneumonia (UIP) and pathological fibroblastic activity [1]. UIP is spatially heterogeneous, both macroscopically and microscopically, with a peripheral and basal predominant distribution. In a CT scan, visible patterns of UIP include honeycombing cysts, reticular opacities associated with traction bronchiectasis, and ground glass opacities [2,3,4]. The presence of any UIP pattern in a CT scan is crucial for IPF diagnosis, and it typically involves a multi-disciplinary team [2,3,4,5]. Semi-quantitative disease severity scoring methods have been proposed that have some prognostic capabilities [6,7]. However, these methods are not currently standardised, and scoring can be subjective across independent radiologists [8].

Several texture-based or machine learning algorithms [9,10,11,12,13,14,15,16] have been proposed for the automated characterisation of CT scans for interstitial lung disease (ILD) patterns that demonstrate correlation and agreement with radiologists’ scoring. Computer-Aided Lung Informatics for Pathology Evaluation and Rating (CALIPER), an image analysis software developed by the Mayo Clinic (Rochester, MN, USA), can automatically characterise and quantify volumetric CT images for patterns of ILD on a voxel-wise level [12]. CALIPER-derived parameters have been shown to be associated with IPF disease progression [17], and were more accurate than visual CT scoring in IPF mortality prediction and prognostication [18,19].

Hyperpolarised gas diffusion-weighted MRI (DW-MRI) with inhaled helium-3 (^3^He) or xenon-129 (^129^Xe) is an imaging technique that is sensitive to changes in acinar microstructure [20,21,22,23]. In lungs with IPF, the global apparent diffusion coefficient (ADC) and mean acinar dimension (Lm_D_) from ^3^He and ^129^Xe DW-MRI is elevated compared to healthy lungs, which is indicative of a loss of the acinar integrity related to fibrosis [24,25,26,27]. Furthermore, DW-MRI metrics correlate with a visual scoring of ILD severity on CT images, and the Lm_D_ demonstrated sensitivity to longitudinal change in IPF [26]. Elevated DW-MRI metric regions qualitatively appeared to spatially correlate with the ILD patterns visible in CT scans, and they were hypothesised to be related to regions of honeycomb cysts. A more regional or voxel-wise comparison is therefore required to help elucidate which ILD features directly correspond to the observed elevated DW-MRI metrics.

Multi-modality spatial co-registration of hyperpolarised gas lung MRI and CT has previously been successfully implemented to compare hyperpolarised gas MRI- and CT-based maps of lung ventilation in patients with asthma [28], chronic obstructive lung disease (COPD) [29] and lung cancer [30]. Moreover, the co-registration of hyperpolarised gas DW-MRI with CT has facilitated quantitative multi-parametric response mapping and MRI-based emphysema indices, which have revealed subclinical features of COPD that were not detectable with DW-MRI or CT alone [31,32,33]. However, to date, there have been no studies that have spatially co-registered DW-MRI and CT in patients with IPF or ILD. The aim of this work was therefore to develop a multi-modality spatial co-registration framework for hyperpolarised gas DW-MRI and CALIPER CT. The framework will facilitate a voxel-wise comparison of DW-MRI metrics with quantitative CALIPER CT patterns in a cohort of IPF patients.

## 2. Materials and Methods

### 2.1. Study Participants

Sixteen patients (mean 71 ± 5 years, 14 men) with a multi-disciplinary team IPF diagnosis, and six healthy volunteers (mean 67 ± 3 years, 4 men) with no history of respiratory disorders and smoking were recruited for this retrospective interpretation of prospectively acquired data from two separate studies that were approved by the Liverpool Central NHS Research Ethics Committee [26] (February 2016 to February 2018) and the Regional Ethical Review Board in Lund, Sweden [34] (March to May 2019), respectively. All participants provided written informed consent.

The inclusion criteria for the patients with IPF included a diagnosis of IPF within one year, oxygen saturations of ≥90% in room air, and an age of 18–80. The exclusion criteria included patients on immunosuppressive treatment, pregnancy, renal impairment, oxygen saturations of <90% in room air, an age of >80 years old (or an age of <18 years old at the onset of the study), an inability to lie supine comfortably for at least 60 min, a significant co-morbidity that was likely to reduce life expectancy to less than one year, a severe ischaemic heart disease (or symptoms of angina that could not be fully controlled), significant congestive cardiac failure, any contraindication(s) to MRI scanning, and previous allergies to MRI contrast agent (gadolinium).

All IPF patients underwent ^3^He MRI and CT at baseline, and eleven patients had a 1-year follow up ^3^He MRI. Four IPF patients died between the follow up examinations, and one patient was too sick and withdrew from the study. All healthy volunteers underwent a baseline ^129^Xe MRI only. The difference in hyperpolarised gas between the IPF patients and healthy volunteers was due to the transition of the research community from ^3^He to ^129^Xe gas due to the scarcity of ^3^He gas [35]. Figure 1 summarises the participant imaging data and analyses for this study.

### 2.2. MRI and CT Image Acquisition

Hyperpolarised ^3^He and ^129^Xe lung MRI was acquired on a 1.5 T GE HDx scanner using ^3^He and ^129^Xe flexible quadrature chest radiofrequency coils (Clinical MR Solutions, Brookfield, WI, USA). All examinations involved the inhalation of a gas mixture of hyperpolarised ^3^He (~25% polarization) or ^129^Xe (~25% polarization), as well as the nitrogen from functional residual capacity (FRC). Gases were polarized under Medicines & Healthcare products Regulatory Agency (MHRA) approved licences with in-house equipment and processes (POLARIS, University of Sheffield, UK). The volume of gas mixture was titrated based upon the subjects’ heights, up to 1 L, to account for the differences in lung volume. Each subject was ≥ 160 cm and subsequently inhaled 1 L gas mixtures. Before undergoing the MRI exam, each subject was coached by a lung physiologist to achieve FRC, and they practiced by inhaling 1 L of room air. Each IPF participant underwent ^3^He DW-MRI and ventilation MRI, while healthy volunteers underwent ^129^Xe DW-MRI only. The aforementioned ^129^Xe and ^3^He DW-MRI sequences were optimised such that comparable Lm_D_ values could be derived from both gases [36].

Hyperpolarised ^3^He diffusion-weighted (DW)-MRI was acquired with a 3D multiple b-value spoiled gradient echo (SPGR) sequence and compressed sensing undersampling [26], which were obtained with the following parameters: 250 mL of ^3^He (balanced with 750 mL of N_2_); FOV: 40 × 32 × 28.8 cm^3^; TE/TR: 4.2/6.0 ms; voxel size: 4.17 × 4.17 × 12 mm^3^; b-values: 0, 1.6, 4.2, and 7.2 s/cm^2^; diffusion time: 1.6 ms; maximum diffusion-weighted gradient strength: 30 mT/m; ramp: 0.3 ms; plateau: 1.0 ms; flip angle: 1.9°; and bandwidth: ±31.25 kHz.

Healthy volunteers underwent hyperpolarised ^129^Xe DW-MRI only with a 3D multiple b-value SPGR sequence and compressed sensing undersampling [36], and these were performed with the following parameters: 550 mL of ^129^Xe (balanced with 450 mL of N_2_); FOV: 40 × 32 × 27 cm^3^; TE/TR: 14.0/17.3 ms; voxel size: 6.25 × 6.25 × 15 mm^3^; b-values: 0, 12, 20, and 30 s/cm^2^; diffusion time: 8.5 ms; maximum diffusion-weighted gradient strength: 32.6 mT/m; ramp: 0.3 ms; plateau: 2.3 ms; flip angle: 3.1°; and bandwidth: ±6.97 kHz. A previous benchmarking study has demonstrated that equivalent ^129^Xe and ^3^He Lm_D_ values can be derived from using the above sequence parameters (see Appendix A) [36].

Hyperpolarised ^3^He lung ventilation MRI was acquired with a 3D balanced steady-state free precession sequence [28] with the following sequence parameters: 150 mL of ^3^He (balanced with 850 mL N_2_); in-plane FOV: 40 × 32 cm^2^; TE/TR: 0.6/1.9 ms; voxel size of 4 × 4 × 5 mm^3^; flip angle: 10°; and bandwidth: ±83.5 kHz. The same-breath ^1^H images of the thorax were acquired at the same spatial resolution as the ^3^He lung ventilation for anatomical reference [28] with the following sequence parameters: 3D SPGR, in-plane FOV: 40 × 40 cm^2^; TE/TR: 0.6/1.4 ms; voxel size of 4 × 4 × 5 mm^3^; flip angle: 5°; and bandwidth: ±83.5 kHz.

The sixteen IPF patients underwent non-contrast multi-detector row CT of the thorax at one tertiary centre on a 64-section scanner (Light-Speed; GE Medical) during a single full-inspiration breath hold. The CT images were reconstructed to 1.25 mm thick sections using either “Soft”, “Lung”, or “Chest” reconstruction kernels, and the mean dose-length product for all participants was 313 mGy∙cm (range, 101–743 mGy∙cm). CT was performed as close as was practical to the MRI examination (mean 56 ± 62 days).

### 2.3. Image Registration and Analysis

The undersampled hyperpolarised ^3^He and ^129^Xe DW-MRI data were reconstructed using in-house MATLAB (MathWorks, Inc., MA, USA) code [26,36]. For each IPF participant, ^3^He DW-MRI was spatially co-registered to the CT indirectly via spatially aligned ^3^He ventilation and structural ^1^H MRI [28], which was achieved using Advanced Normalization Tools (ANTs) software [37] (Figure 2). Furthermore, ^3^He DW-MRI was co-registered to the ^3^He ventilation images, while CT was co-registered to the structural ^1^H MRI. CT images were segmented as part of the CALIPER software analysis; meanwhile, all ^3^He and ^1^H MRI images were segmented using in-house developed software. Each image registration involved a rigid pre-alignment transformation that was followed by affine and diffeomorphic transformations [28]. For the diffeomorphic stage, a standard pyramidal approach was followed using mutual information at the higher levels [38] and cross correlation at the base as cost functions [39]. Further details on the image registration transformations can be found in Tahir et al. [28]. Image registrations were assessed by Dice similarity coefficients between the binary lung segmentation masks of warped ^3^He DW-MRI and ^3^He ventilation, as well as with warped CT and structural ^1^H.

The CT images were analysed with CALIPER software, wherein each parenchyma voxel was characterised into one of seven patterns: normal, honeycombing, reticular changes, ground-glass opacities, and lower attenuation areas (LAA) (mild, moderate, and severe) [12]. To reduce the number of CALIPER patterns and DW-MRI comparisons, additional patterns were defined. Non-involved represents the physiologically normal lung, as well as the combined normal and mild LAA patterns; this was such because mild LAA patterns can appear in healthy lungs after deep inhalation. Hyperlucent voxels represent the emphysematous regions of the lung (moderate or severe LAA patterns). Thus, maps containing five CALIPER patterns (non-involved, honeycombing, reticular changes, ground-glass opacities, and hyperlucent) were derived for each CT image set.

CALIPER maps were co-registered to DW-MRI using the same CT-^1^H deformation field transformation with the nearest neighbour interpolation. Then, ^3^He ADC and Lm_D_ values were calculated for each original IPF ^3^He DW-MRI dataset on a voxel-by-voxel basis. Moreover, the ^3^He ADC was calculated from a mono-exponential fit of two b-values (b = 0, and 1.6 s/cm^2^). Furthermore, ^3^He Lm_D_ was derived from fitting all respective 4 b-values (^3^He b = 0, 1.6, 4.2, 7.2 s/cm^2^) to the stretched exponential model (SEM) of the gas diffusion in the lungs [34]. Voxel-wise maps of the ^3^He ADC and Lm_D_ were subsequently warped using the ^3^He DW-MRI-ventilation deformation field transformation with the nearest neighbour interpolation. For each healthy volunteer, maps of ^129^Xe Lm_D_ were calculated voxel-by-voxel from the ^129^Xe DW-MRI data using the SEM for ^129^Xe b= 0, 12, 20, and 30 s/cm^2^ [34].

In the absence of longitudinal CT imaging, the 1-year follow up of ^3^He DW-MRI, available for eleven of the IPF patients, was warped to the spatial domain of the baseline ^3^He ventilation using the same ANT registration pipeline detailed above for the baseline ^3^He DW-MRI. Thus, after this additional registration step, both baseline and longitudinal ^3^He DW-MRI were spatially co-registered to the baseline CT and CALIPER maps.

### 2.4. Statistical Analyses

The overlapping voxels from spatially co-registered ^3^He ADC or Lm_D_ maps and CALIPER maps were compared across all of the IPF patients. Only 5% of the overlapping voxels (every 20th) were considered for statistical analyses. This was due to the computational limitations of the statistical analysis software because of the large (~4 million) number of overlapping voxels.

Statistical differences in the ^3^He ADC or Lm_D_ values between the five CALIPER patterns were assessed with one-way ANOVA and the post hoc Tukey multiple comparison tests. The IPF patients’ ^3^He Lm_D_ values in CALIPER non-involved pattern voxels were compared to the healthy volunteers’ ^129^Xe Lm_D_ values with an independent t-test. A ^129^Xe Lm_D_ threshold of 406 µm, corresponding to the 95% upper limit of healthy ^129^Xe Lm_D_ values (see Results), was used to classify the ^3^He Lm_D_ voxels in the IPF cohort that were greater than the threshold as an abnormal value. The percentage of lung voxels classified as abnormal by Lm_D_ was compared to those co-registered CALIPER voxels with ILD patterns (honeycombing, reticular changes, and ground-glass opacities) using Bland–Altman analysis. In the sub-cohort of eleven IPF patients with longitudinal ^3^He DW-MRI, the overlapping voxels in ^3^He ADC or Lm_D_ and CALIPER maps were compared at baseline and after 1 year. Independent t-tests for the overlapping voxels in each of the five CALIPER patterns were used to determine if the statistically significant longitudinal changes in the diffusion metrics were observed in each respective CALIPER pattern.

All statistical analyses were performed in GraphPad Prism (v9.5, La Jolla, CA, USA), and any *p*-values that were <0.05 indicated statistical significance. Any statistically significant difference in ADC or Lm_D_ values were compared to a respective a priori-defined confidence interval (CI) range to contextualise if the statistical difference was a relevant one. The mean difference 95% CI of the Tukey multiple comparison tests or independent t-test differences were compared to the Bland–Altman 95% difference interval range that was previously reported for the same-day reproducibility of ^3^He ADC (±0.041 cm^2^/s) and Lm_D_ (±18.5µm) values in IPF patients [26].

## 3. Results

### 3.1. Spatial Co-Registration

Table 1 summarises the demographics and DW-MRI metrics for the IPF patients and healthy volunteers at baseline. The spatial co-registration of CT and ^3^He DW-MRI was successfully implemented in all sixteen IPF patients, and the resultant spatial resolution of co-registered images was 1.8 × 1.8 × 5 mm^3^. The mean Dice similarity coefficient for the CT-^3^He DW-MRI spatial co-registration was 0.920 ± 0.013. The mean Dice coefficients for the two separate CT-structural ^1^H and ^3^He DW-MRI-ventilation co-registrations were 0.954 ± 0.008 and 0.922 ± 0.009, respectively. The individual IPF patients’ co-registration Dice coefficients, CALIPER pattern percentages, and ^3^He DW-MRI metrics are summarised in Appendix A.

### 3.2. Baseline Voxel Comparison

An analysis of the overlapping co-registered voxels that were present in all baseline IPF patients demonstrated that the voxels classified as honeycombing had the largest ^3^He ADC and Lm_D_ values for all CALIPER patterns (Table 2).

One-way ANOVA revealed that there was a statistically significant difference in the ^3^He ADC value (F(4, 203670) = 2618, *p* < 0.001) between at least two CALIPER patterns in the overlapping voxels (Figure 3a). The post hoc Tukey tests for multiple comparisons found significantly different ^3^He ADC values between all five CALIPER patterns (*p* < 0.001) (Table 3). When the 95% CI of the differences were compared to the a priori-defined relevance range, the mean difference in the ^3^He ADC between all CALIPER patterns was relevant, except that between the reticular and ground-glass patterns (Figure 3b).

Similar trends were observed for ^3^He Lm_D_, where one-way ANOVA indicated statistically significant differences in the CALIPER patterns (F(4, 186218) = 665.9, *p* < 0.001) (Figure 4a). The post hoc Tukey tests for multiple comparisons found significantly different ^3^He Lm_D_ values between all five CALIPER patterns (*p* < 0.001), except between the reticular and ground-glass (*p* = 0.87), hyperlucent and ground-glass (*p* = 0.16), and hyperlucent and reticular (*p* = 0.051) patterns (Table 3). All statistically significant ^3^He Lm_D_ differences in the CALIPER patterns were relevant when compared against the a priori ^3^He Lm_D_ relevance range (Figure 4b).

The ^3^He Lm_D_ values within the non-involved CALIPER pattern (mean = 380 ± 87 µm) were significantly larger (t(190731) = 168.7, *p* < 0.001) than the older healthy volunteer ^129^Xe Lm_D_ values (mean = 300 ± 61 µm) (Figure 4a). A threshold of 406 µm, corresponding to the 95% upper limit of older healthy ^129^Xe Lm_D_ values was chosen to classify the ^3^He Lm_D_ maps for abnormality. A Bland–Altman analysis of the percentage of abnormal lung voxels between Lm_D_ (>406 µm) and CALIPER (all ILD patterns) classifications obtained a mean bias of −15.3% (95% confidence interval −56.8% to 26.2%) towards CALIPER for IPF participants (Figure 5a). These results suggested that more abnormal microstructural changes are detected by DW-MRI. A trend towards increasing bias between CALIPER and Lm_D_ with an increased percentage of abnormal voxels was also observed (Figure 5b).

### 3.3. Longitudinal Voxel Comparison

For the sub-cohort of 11 IPF patients who underwent a 1-year follow-up DW-MRI, overlapping co-registered 1-year ADC or Lm_D_ voxels were statistically significant different (*p* < 0.001) after 1 year globally and in all baseline CALIPER patterns, except in the hyperlucent and honeycomb patterns for ADC, and in the honeycomb pattern for Lm_D_ (Table 4). When significant longitudinal differences in CALIPER patterns were considered against the a priori ^3^He ADC and Lm_D_ relevance ranges, the largest and only relevant differences were observed for reticular (ADC and Lm_D_) and ground-glass patterns (ADC). Meanwhile, the changes in non-involved and hyperlucent patterns were not relevant (Figure 6). This suggests that the longitudinal DW-MRI changes observed in this IPF cohort occur in the regions of the lung with ILD patterns, and not due to increased emphysematous regions.

## 4. Discussion

A framework for the spatial co-registration of ^3^He DW-MRI and CT images was developed and implemented using images from the sixteen IPF participants. A high Dice similarity coefficient for the resultant spatial transformation indicates an excellent spatial overlap of the ^3^He DW-MRI and CT imaging modalities. The choice of an indirect CT and DW-MRI registration framework using spatially aligned ^3^He-^1^H MRI was based on a previous study of CT and ^3^He ventilation MRI registration, which demonstrated more accurate registrations with an indirect method that utilised same-breath ^3^He-^1^H MRI than a direct registration method [28]. The slightly lower Dice coefficient for the ^3^He DW-MRI-ventilation transformation may be related to the inherently lower spatial resolution of DW-MRI, which can result in fewer ventilation defects that are visible on the respective images and binary segmentation masks.

The CALIPER honeycombing pattern voxels had the largest ADC and Lm_D_ values out of all the CALIPER patterns; this further supports the hypothesis that elevated hyperpolarised gas DW-MRI metrics are a result of honeycomb cysts [26]. Smaller differences in the ADC and Lm_D_ values between normal and reticular or ground-glass patterns, when compared to honeycombing, may suggest that, in a CT scan, these ILD patterns have minimal accompanying acinar microstructural changes. However, the Lm_D_ values in the CALIPER non-involved or physiologically normal patterns were also significantly larger than those obtained from the healthy volunteers of a similar age range. This would suggest DW-MRI may detect microstructural changes in the regions of the IPF lung that quantitative CT characterises as physiologically normal. When the abnormal Lm_D_ threshold, defined as the 95% upper limit of healthy volunteer ^129^Xe Lm_D_ values, was compared to the CALIPER ILD patterns in IPF patients, an overall bias towards a higher percentage of the lung being classified as abnormal with DW-MRI was obtained. This bias further suggests that hyperpolarised gas DW-MRI may be sensitive to aspects of acinar changes in IPF lung disease, such as microscopic cysts that are not resolved by CT.

In our IPF cohort, the percentage of lung voxels characterised as ground-glass opacities by CALIPER was relatively high (12.4%, Table 2), which is not typical for a radiologic UIP pattern in a CT scan. However, we can confirm that each IPF subject had either a definite UIP or probable UIP CT pattern, as visually assessed by thoracic radiologists during multi-disciplinary team diagnosis (see Appendix A), and this is indicative of no predominant regions of ground-glass opacities [2,3,4]. This discrepancy in ground-glass opacity classifications could be related to differences between CALIPER and radiologist visual scoring, in which it has been previously shown that the honeycombing regions identified by radiologists were frequently characterised as reticular changes or ground-glass opacities on CALIPER [19].

A trend towards increased mean global ADC and Lm_D_ values in spatially co-registered 1-year follow ups of ^3^He DW-MRI is in keeping with the trends observed in the wider IPF patient cohort of this study [26]. The largest increases in ADC and Lm_D_ voxels after 1 year were observed for the patterns of reticular changes and ground-glass opacities, and not due to emphysema. These results suggest that if 1-year follow-up CALIPER CT images were acquired, then they would show an increased lung percentage of CALIPER ILD patterns [18,19]. We also hypothesise that some baseline ground-glass and reticular patterns regions would be classified as honeycombing in a 1-year follow-up CT scan.

The main limitation was that our IPF patient cohort was small and no longitudinal CT imaging was available. More patient data, across the two modalities, are therefore required to confirm our baseline and longitudinal findings. There were also limitations in the voxel-wise comparison of CALIPER and DW-MRI. First, due to the relatively large differences in voxel resolution between DW-MRI and CT, DW-MRI voxels were up-sampled and interpolated during spatial co-registration. The combination of partial volume voxel effects and misalignment errors may result in the incorrect classification of DW-MRI voxels in the regions with subtle changes in CALIPER CT patterns. However, the large number of voxels in the comparison, albeit in a small cohort, is a strength of this study and helps minimise the possibility of registration errors affecting our voxel-wise comparison. Second, only overlapping co-registered image voxels were considered. DW-MRI metrics are derived from ventilated lung voxels only; therefore, a comparison was not possible in ventilation defect regions. This is, however, more relevant in obstructive lung diseases when compared to restrictive ones such as IPF, and this is supported by the small number of un-ventilated lung regions that were observed on ^3^He DW-MRI.

The differences in imaging protocols between IPF patients and healthy volunteers, as well as between CT and DW-MRI, are also limitations of this study. The difference in hyperpolarised gas for DW-MRI was mitigated by implementing optimised diffusion imaging parameters that result in comparable DW-MRI metrics (see Appendix A) [36]. However, the inherent lower diffusivity of ^129^Xe gas may slightly underestimate the healthy volunteers’ Lm_D_ values that are in the upper limit of normal when compared to ^3^He gas. The CT scans were acquired at full inspiratory volume, while ^3^He DW-MRI was imaged at FRC + 1L. Therefore, lung inflation volume differences could be a factor because DW-MRI metrics are more homogeneous and larger at full inspiration [40], and consequently the threshold for abnormal Lm_D_ values may be underestimated due to the smaller Lm_D_ values in the dependent lung at FRC + 1L. The combination of inflation and diffusivity differences may lead to an overestimation of the true bias between DW-MRI and CT if both imaging modalities were acquired at the same inflation level and with ^3^He gas.

## 5. Conclusions

The spatial co-registration of hyperpolarised gas DW-MRI and CALIPER quantitative CT maps in IPF patients demonstrated that the largest ADC and Lm_D_ values were observed in the regions classified as honeycombing. In addition, longitudinal DW-MRI changes were predominantly observed in reticular change and ground-glass opacity regions. Furthermore, the Lm_D_ values in voxels with a CALIPER normal pattern were larger than those from age-matched healthy volunteers, thus suggesting DW-MRI may detect microstructural changes even in areas of the lung that are determined as structurally normal by CT scans. The quantitative biomarkers from hyperpolarised gas DW-MRI and CT could play a role in future clinical trials, whereby IPF disease progression and response to new treatments is assessed. With the transition from hyperpolarised ^3^He to ^129^Xe for clinical lung imaging studies [35], this spatial co-registration framework is immediately transferrable to ^129^Xe DW-MRI. Furthermore, it can be used to explore quantitative CT patterns in different ILD subtypes and pulmonary diseases.

## Figures and Tables

**Figure 1 diagnostics-13-03497-f001:**
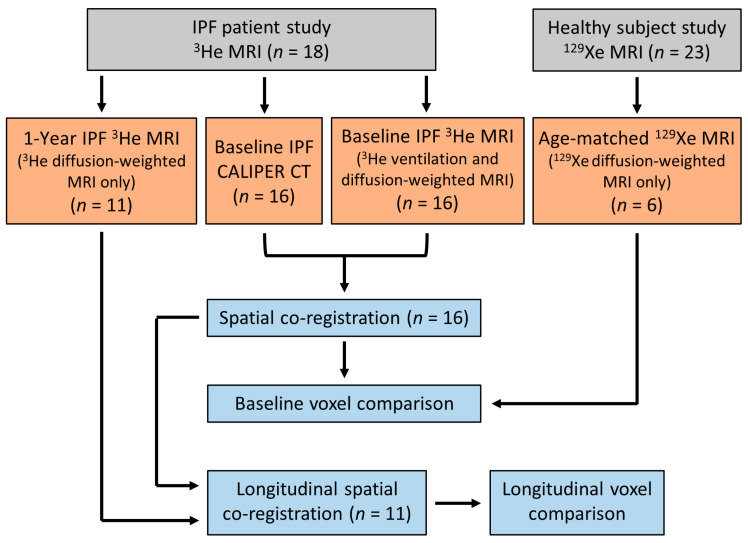
Flow chart summarising the imaging data obtained and the imaging analyses for this study. IPF patient and healthy volunteer data were from two separate prospective studies, respectively [26,34].

**Figure 2 diagnostics-13-03497-f002:**
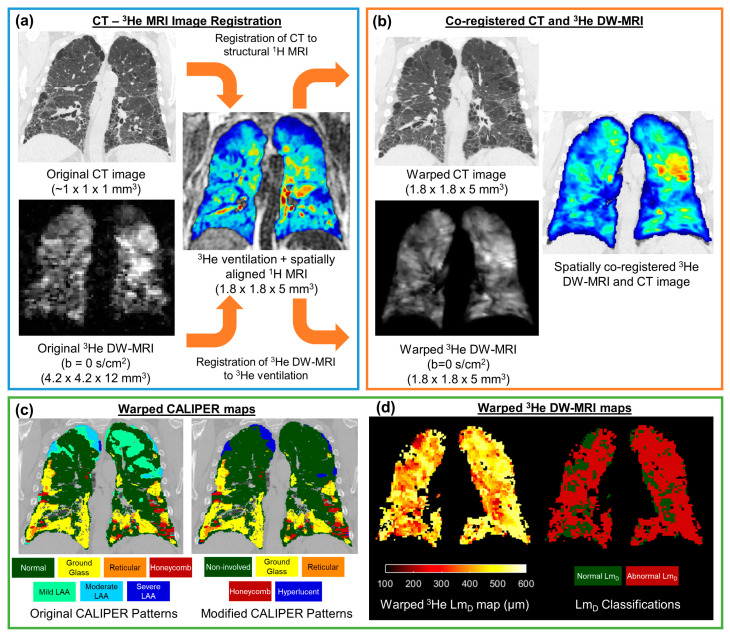
Framework for the spatial co-registration of CT images and ^3^He diffusion-weighted (DW)-MRI. The original CT images and DW-MRI (**a**) were indirectly co-registered by utilising same-breath acquired ^3^He ventilation and structural ^1^H MRI (**b**). The resultant warped images had a spatial resolution of 1.8 × 1.8 × 5 mm^3^ (**c**). The spatial transformation used to warp CT images was also used to deform CALIPER classifications maps. (**d**) Maps of the ^3^He ADC and Lm_D_ were calculated from the original ^3^He DW-MRI, and they were warped using the same spatial transformation as DW-MRI. The abnormal Lm_D_ values were defined from a threshold of 406 µm, and they were derived from older healthy volunteer ^129^Xe DW-MRIs.

**Figure 3 diagnostics-13-03497-f003:**
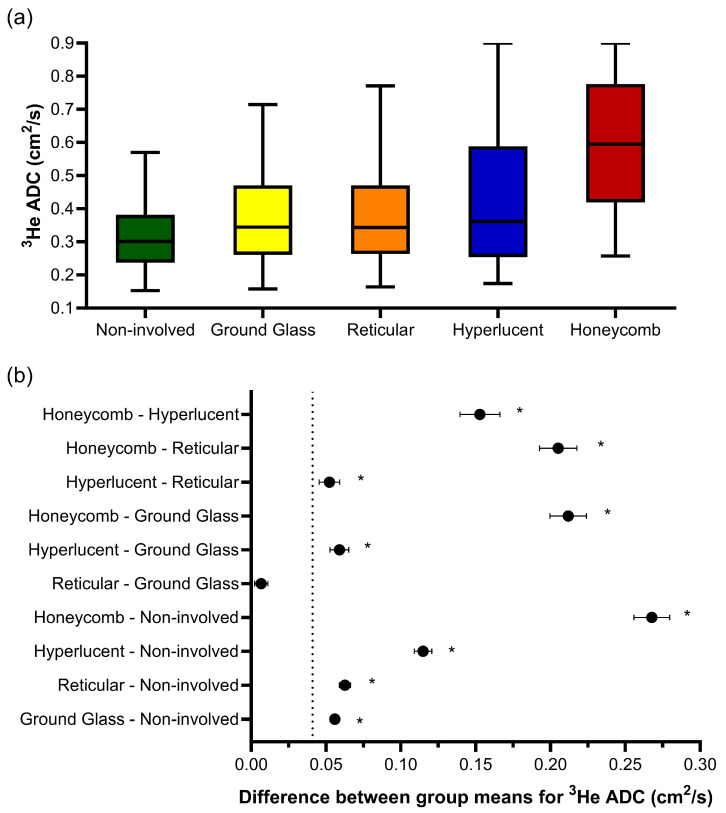
(**a**) Boxplots of the ^3^He ADC value for each CALIPER pattern from a voxel-wise comparison of all the overlapping co-registered voxels in all IPF patients. The boxplot whiskers are representative of 5th and 95th percentiles. A significant difference between the CALIPER patterns for ^3^He ADC was obtained with a one-way ANOVA test (*p* < 0.001). (**b**) Plots of the mean difference confidence intervals (CI) for each post hoc Tukey multiple comparison test. The comparisons denoted by asterisks were significantly different (*p* < 0.001) and had a mean difference CI that was greater than the a priori-defined relevance range for ^3^He ADC in IPF patients (±0.041 cm^2^/s, dotted line).

**Figure 4 diagnostics-13-03497-f004:**
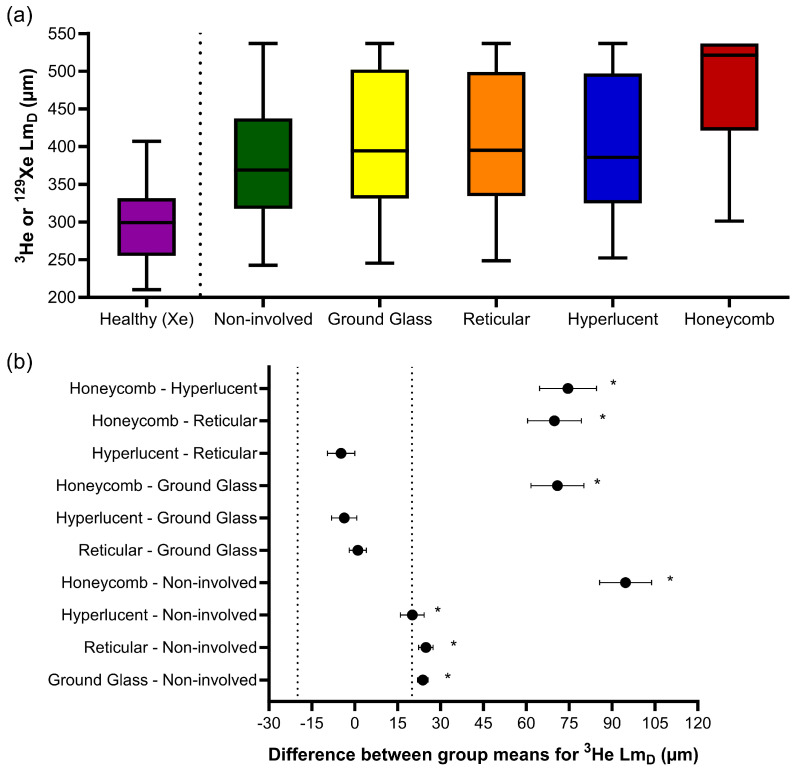
(**a**) Boxplots of the ^3^He Lm_D_ values for each CALIPER pattern obtained via a voxel-wise comparison of all the overlapping co-registered voxels in all IPF patients, and the ^129^Xe Lm_D_ values for the older healthy volunteers. Boxplot whiskers are representative of the 5th and 95th percentiles. A significant difference between the CALIPER patterns for ^3^He Lm_D_ was obtained with a one-way ANOVA test (*p* < 0.001). The non-involved CALIPER patterns’ ^3^He Lm_D_ was significantly larger (*p* < 0.001, +80.1 µm) than those in the older healthy ^129^Xe Lm_D_ values. (**b**) The plots of mean difference confidence intervals (CI) for each post hoc Tukey multiple comparison tests. The comparisons denoted by asterisks were significantly different (*p* < 0.001) and had a mean difference CI greater than the a priori-defined relevance range for ^3^He Lm_D_ in the IPF patients (±18.5 µm, dotted line).

**Figure 5 diagnostics-13-03497-f005:**
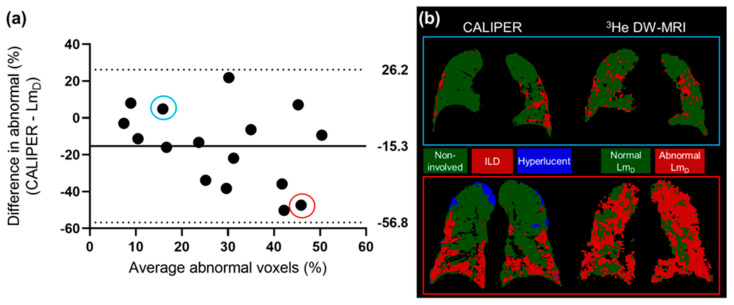
(**a**) The Bland–Altman analysis of the percentage of voxels classified as abnormal by CALIPER or Lm_D_ (>406 µm) in all of the IPF patients. A mean bias of −15.3% towards abnormal values as classified by CALIPER was observed. (**b**) Two example patients with IPF, where one patient demonstrates a small difference in the percentage of their lungs classified as abnormal (blue—CALIPER = 18.3%, Lm_D_ = 13.4%), and one patient demonstrates a large difference (red—CALIPER = 22.2%, Lm_D_ = 69.6%).

**Figure 6 diagnostics-13-03497-f006:**
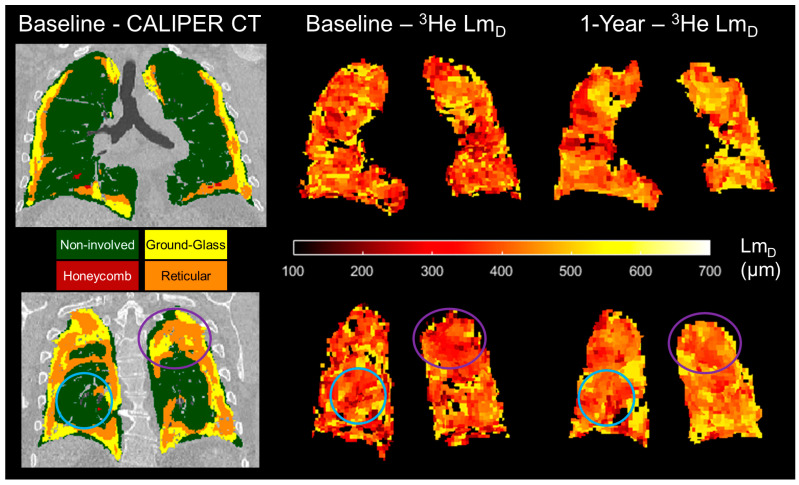
Two example spatially co-registered ^3^He Lm_D_ maps at baseline and after 1 year following the corresponding baseline CALIPER CT map in one representative IPF patient. In the regions classified as ground-glass and reticular, the largest longitudinal differences in the ^3^He Lm_D_ value were observed (purple circles). In contrast, the regions that were classified as CALIPER non-involved demonstrated much less of a change in the ^3^He Lm_D_ value after 1 year (blue circles).

**Table 1 diagnostics-13-03497-t001:** A summary of the subject demographics and global DW-MRI metrics for IPF patients and older healthy volunteers. All values are given as the mean ± standard deviation.

	IPF	Healthy Volunteer
No. of subjects	16	6
Sex	14M, 2F	4M, 2F
Age (years)	70.6 ± 5.2	66.7 ± 2.8
FEV_1_ (% predicted)	79.2 ± 12.7	98.6 ± 8.8
FVC (% predicted)	79.9 ± 17.9	99.1 ± 11.8
D_LCO_ (% predicted)	48.3 ± 20.9	108.1 ± 19.5
Global ADC (cm^2^/s) *	0.335 ± 0.075	0.038 ± 0.004
Global Lm_D_ (µm) *	384 ± 41	299 ± 19

* ^3^He DW-MRI was acquired in the IPF cohort, and ^129^Xe DW-MRI was in the healthy volunteer cohort.

**Table 2 diagnostics-13-03497-t002:** A summary of ^3^He ADC and Lm_D_ values for each CALIPER pattern in the baseline voxel-wise comparison of the 16 IPF patients. All values are given as the mean ± standard deviation.

CALIPER Pattern	% of Voxels	^3^He ADC (cm^2^/s)	^3^He Lm_D_ (µm)
Non-involved	78.9	0.323 ± 0.133	380 ± 89
Ground-glass	12.4	0.379 ± 0.169	404 ± 98
Reticular	5.5	0.385 ± 0.179	405 ± 96
Hyperlucent	2.7	0.437 ± 0.235	400 ± 95
Honeycomb	0.5	0.588 ± 0.213	473 ± 82

**Table 3 diagnostics-13-03497-t003:** Summary of mean differences (95% confidence interval range) for the post hoc Tukey multiple comparison tests of the ^3^He ADC and Lm_D_ with CALIPER patterns.

Post hoc Tukey Tests		Mean Difference ^3^He ADC (cm^2^/s) (Column–Row)				
		Non-Involved	Ground-Glass	Reticular	Hyperlucent	Honeycomb
**Mean difference ^3^He Lm_D_ (µm) (row–column)**	**Non-involved**	-	0.056 *^#^ (0.053, 0.059)	0.063 *^#^ (0.059, 0.066)	0.115 *^#^ (0.109, 0.121)	0.268 *^#^ (0.256, 0.280)
	**Ground-glass**	23.8 *^#^ (22.0, 25.7)	-	0.007 * ^nr^ (0.002, 0.011)	0.059 *^#^ (0.053, 0.065)	0.212 *^#^ (0.200, 0.224)
	**Reticular**	24.9 *^#^ (22.4, 27.4)	1.1 (−1.9, 4.1)	-	0.052 *^#^ (0.045, 0.059)	0.205 *^#^ (0.193, 0.218)
	**Hyperlucent**	20.1 *^#^ (16.0, 24.3)	−3.7 (−8.1, 0.7)	−4.8 (−9.5, 0.0)	-	0.153 *^#^ (0.140, 0.166)
	**Honeycomb**	94.7 *^#^ (85.6, 103.8)	70.9 *^#^ (61.7, 80.2)	69.9 *^#^ (60.4, 79.3)	74.6 *^#^ (64.6, 84.6)	-

* = significant to the *p* < 0.001 level; # = relevant, and the mean difference 95% confidence interval range is larger than the respective relevance range for the ^3^He ADC (±0.041 cm^2^/s) and Lm_D_ (±18.5 µm) values; and nr = not relevant.

**Table 4 diagnostics-13-03497-t004:** Summary of the baseline and 1-year follow up ^3^He DW-MRI metrics in the longitudinal sub-cohort of the 11 IPF patients. Values given as the mean ± standard deviation. The mean differences (95% confidence interval range) for each CALIPER pattern, as classified on baseline CT, are shown for the ^3^He ADC and Lm_D_ values obtained from independent t-tests.

	Baseline		1 Year		Mean Difference (1 Year—Baseline)	
	^3^He ADC (cm^2^/s)	^3^He Lm_D_ (µm)	^3^He ADC (cm^2^/s)	^3^He Lm_D_ (µm)	^3^He ADC (cm^2^/s)	^3^He Lm_D_ (µm)
Non-involved	0.313 ± 0.124	375 ± 88	0.315 ± 0.115	380 ± 81	0.002 * ^nr^ (0.001, 0.003)	5.1 * ^nr^ (4.4, 5.8)
Ground-glass	0.358 ± 0.163	391 ± 97	0.387 ± 0.168	411 ± 92	0.029 * ^nr^ (0.025, 0.033)	19.8 *^#^ (17.2, 22.4)
Reticular	0.337 ± 0.138	389 ± 89	0.375 ± 0.143	414 ± 84	0.038 *^#^ (0.033, 0.043)	25.2 *^#^ (22.0, 28.4)
Hyperlucent	0.396 ± 0.213	389 ± 92	0.397 ± 0.205	397 ± 84	0.001 (−0.008, 0.010)	7.3 * ^nr^ (3.2, 11.5)
Honeycomb	0.595 ± 0.218	472 ± 85	0.622 ± 0.220	476 ± 78	0.027 (−0.007, 0.062)	4.2 (−11.7, 20.2)
Global	0.320 ± 0.134	377 ± 90	0.326 ± 0.128	385 ± 83	0.005 * ^nr^ (0.004, 0.006)	7.4 * ^nr^ (6.8, 8.1)

* = significant to the *p* < 0.001 level; ^#^ = relevant, and the mean difference 95% confidence interval range is larger than the respective relevance range for the ^3^He ADC (±0.041 cm^2^/s) and Lm_D_ (±18.5 µm) values; and nr = not relevant.

## Data Availability

The data presented in this study are available on request from the corresponding author.

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
