# Peer review of "Voxel-Wise Comparison of Co-Registered Quantitative CT and Hyperpolarised Gas Diffusion-Weighted MRI Measurements in IPF"

_diagnostics, 2023, doi:10.3390/diagnostics13233497_

Round 1
Reviewer 1 Report
Thank you for submitting this interesting study.
Author Response
We would like to thank the reviewer for dedicating their time to review our manuscript.
Reviewer 2 Report
I would like to thank the handling editor for offering me the opportunity to review the manuscript entitled “Voxel-wise comparison of co-registered quantitative CT and hyperpolarised gas diffusion-weighted MRI measurements in IPF” authored by Chan and colleagues, which is currently under consideration for publication in Diagnostics. I would also like to commend the authors for their scholarly work, which investigates the correlation between lung imaging biomarkers from hyperpolarized gas diffusion-weighted MRI (DW-MRI) and quantitative CT analysis in patients with idiopathic pulmonary fibrosis (IPF). The authors developed a framework to spatially co-register DW-MRI and CT scans in 16 IPF patients. Voxel-wise comparison between DW-MRI metrics (apparent diffusion coefficient and mean acinar dimension) and CALIPER CT patterns was performed. The results indicate the largest DW-MRI values were found in honeycombing regions on CT. DW-MRI metrics in CT-defined normal regions were higher than healthy controls, suggesting MRI detects microstructural changes missed by CT. Longitudinal DW-MRI changes occurred predominantly in ground-glass and reticular CT patterns. The manuscript provides initial evidence that combining DW-MRI and quantitative CT could improve assessment of lung microstructure in IPF. The authors claim DW-MRI may identify pathological changes in areas of the lung appearing normal on CT. Further research with more patients and longitudinal CT data is required to confirm the findings. Overall, the study demonstrates the potential value of integrating DW-MRI and CT biomarkers for IPF evaluation.
The manuscript under review appears to provide some novel contributions to the existing literature on imaging biomarkers in idiopathic pulmonary fibrosis. A key strength of the study is the development of a spatial co-registration framework to enable voxel-wise comparison between diffusion-weighted MRI and quantitative CT measurements in the lungs. To my knowledge, this is the first application of this multi-modality image registration approach in patients with IPF specifically. The voxel-level analysis provides greater regional detail than previous studies that compared global MRI and CT metrics. The findings that MRI detects microstructural changes in areas appearing normal on CT, and that longitudinal MRI changes align with CT patterns of disease progression, are important preliminary results. If validated in larger studies, this suggests diffusion-weighted MRI could identify pathological changes at an earlier stage than CT alone.
Overall, the integration of MRI and CT biomarkers demonstrated in this study represents an advance in the quantitative imaging of IPF microstructure. If published, the manuscript would make a valuable contribution to the literature and could spur further research capitalizing on the complementary information from the two modalities.
While the manuscript provides valuable insights, there are a few areas that could be refined to further augment the quality and impact of the work. Here are some respectful suggestions the authors could consider to potentially improve the manuscript:
Introduction
· Expand the introduction to provide more background on other studies that have analysed or co-registered CT and MRI metrics in interstitial lung diseases besides IPF. This would help frame the novelty of your approach focused specifically on IPF.
Methods
· Provide more details on the image analysis and registration pipeline. For example, explain how lung segmentation was performed and what cost functions were used in the registration. This will aid reproducibility.
· Provide more specifics on the reconstruction kernels. This technical detail may be important for interpreting the CT quantitative analysis.
· Explain if there were any inclusion/exclusion criteria for the IPF patient cohort beyond diagnosis. Characterizing the population will aid generalization.
Results
· Consider analysing sub-regions of the lungs (e.g., apex to base) in addition to whole lung. This could reveal regional variations.
· Include error bars or confidence intervals on the quantitative ROI measurements and graphs to demonstrate the variability.
· Perform correlation analysis between CT patterns/severity and MRI metrics. This would further validate the relationship.
Discussion
· Discuss limitations of indirectly co-registering MRI and CT through ventilation images rather than directly. Are there concerns about deformation mismatches?
· Highlight specific future studies the co-registration framework enables for IPF assessment. This will showcase the impact of your method.
· Discuss challenges of overcoming partial volume effects in the voxel-wise analysis due to resolution differences between MRI and CT.
· Highlight other potential applications of the co-registration framework besides IPF, such as COPD or asthma. This expands the impact.
· Elaborate on the potential clinical utility and workflow for integrated MRI-CT metrics in managing IPF patients. This will emphasize translational value.
In conclusion, I would like to reiterate my appreciation to both the editor and the authors for the opportunity to review this intriguing and informative manuscript. I trust that my suggestions could help enhance the clarity, robustness, and relevance of this important work. I look forward to seeing the revised version of the manuscript and wish the authors success in their ongoing research endeavours.
Reviewer 3 Report
This manuscript by Chan al aims to evaluate if DW-MRI can be used for assessing IPF, and comparing the readouts of DW-MRI to CT with a commercially available software package (CALIPER). The following is a synopsis of concerns raised based on the data presented in the manuscript.
Major:
1) Total lung volume should be provided in Table 1.
2) Were the MRI and CT performed in a prone or supine position? Were the IPF subjects on long-term oxygen therapy? The authors should also provide what CT patterns IPF subjects had. UIP, probable UIP? Indeterminate for UIP? How many of the patients had VATS- biopsy to diagnose UIP?
3) It is interesting that ground glass, which is less common in UIP, represented 12.4% of the voxels, but fibrotic changes (reticulation and honeycombing) only have 6% in total. Were CT and MRI images acquired only during inspiration? It is interesting to see a large area of upper-lobe predominant ground glass in Figure 6, which raised concern about chronic hypersensitivity pneumonitis.
4) It would be ideal if the authors had follow-up CT for comparison.
5) It is unclear why ADC is higher in hyperlucent (emphysema) than ground glass and reticulation.
6) Can the authors postulate which patient population may benefit more from DW-MRI? Given the significant cost differences and risk of radiation exposure associated with DW-MRI.
Round 2
Reviewer 2 Report
I want to express my appreciation for the attention and consideration you have devoted to my suggested revisions for your manuscript. It is evident that a significant amount of effort and thought has been directed towards the refining of your work, integrating the feedback provided during the peer review process. The resulting modifications demonstrate a thorough and thoughtful approach, and significantly enhance the rigor and overall quality of your manuscript. I look forward to witnessing the impact your research will undoubtedly have on the academic community.
Author Response
We would like to thank the reviewer again for dedicating their time to review our revised manuscript
Reviewer 3 Report
Response 1 and 3: The authors should explain how they calculate FRC without lung volume information. Spirometry is insufficient to provide FRC.
Response 2: Image pattern is crucial in IPF diagnosis. The authors should request an extension from the editorial office and provide relevant data.
Response 5: If, as the authors stated, higher ADC/LmD in emphysema is due to microstructure distortion and free diffusion, why do honeycombing changes (end-stage fibrosis, which is known to impair diffusion) have the highest ADC?
Response 6: MRI is rarely done in pediatric patients due to the frequent need for sedation.
Author Response
We thank the reviewer for their time reviewing our revised manuscript. Please see attached for our response to the reviewer's comments

Round 3
Reviewer 3 Report
The authors have addressed my concerns, particularly regarding the groundglass changes detected by CT.